# Probing nanofriction and Aubry-type signatures in a finite self-organized system

J. Kiethe[1], R. Nigmatullin[2,3], D. Kalincev[1], T. Schmirander[1] & T.E. Mehlstäubler[1]

Friction in ordered atomistic layers plays a central role in various nanoscale systems ranging from nanomachines to biological systems. It governs transport properties, wear and dissipation. Defects and incommensurate lattice constants markedly change these properties. Recently, experimental systems have become accessible to probe the dynamics of nanofriction. Here, we present a model system consisting of laser-cooled ions in which nanofriction and transport processes in self-organized systems with back action can be studied with atomic resolution. We show that in a system with local defects resulting in incommensurate layers, there is a transition from sticking to sliding with Aubry-type signatures. We demonstrate spectroscopic measurements of the soft vibrational mode driving this transition and a measurement of the order parameter. We show numerically that both exhibit critical scaling near the transition point. Our studies demonstrate a simple, well-controlled system in which friction in self-organized structures can be studied from classical- to quantum-regimes.

[1] Physikalisch-Technische Bundesanstalt, Bundesallee 100, 38116 Braunschweig, Germany. [2] Complex Systems Research Group, Faculty of Engineering and IT, The University of Sydney, Sydney, New South Wales 2006, Australia. [3] Department of Materials, University of Oxford, Parks Road, Oxford OX1 3PH, UK. Correspondence and requests for materials should be addressed to T.E.M. (email: tanja.mehlstaeubler@ptb.de).

D ry friction is the resistance to the relative movement of two solid layers. It is responsible for many phenomena such as earthquakes, wear or crack propagation and is of enormous practical and technological impact[1]. According to Amontons and Coulomb, friction between solids is proportional to the normal force but independent of the contact areas. This intriguing result was explained by realizing that macroscopic objects touch at asperities that are deformed[2]. A different signature occurs when atomically flat surfaces slide against each other, as for example encountered in micro- or nanoelectro-mechanical systems or biological molecular motors[1,3,4]. At this nanoscale level the friction is no longer described by the Amontons-Coulomb law. For this, mathematical models were developed which are simple enough to be analysed analytically and yet should capture the salient features of the friction phenomena. As the sliding atomic layers are in contact with a thermal environment, dry friction phenomena are a problem of non-equilibrium statistical mechanics as well as nonlinear dynamics[5].

One of the most successful models describing friction phenomena is the Frenkel–Kontorova (FK) model[6]. It consists of a chain of coupled particles sliding over a static periodic potential, which mimics a rigid undeformable substrate. The analysis of this model has revealed highly nontrivial, nonlinear dynamics such as the creation of kinks and anti-kinks, which facilitate the sliding. For an infinite system with incommensurate lattice periodicities, this model displays the celebrated Aubry transition[7], where the sliding motion becomes frictionless, due to the competition of different interaction energies in the atomic many-body system. In solid-state systems, this superlubric regime has been demonstrated in nanocontacts of graphene and gold surfaces[8–11]. In finite systems a smooth-sliding regime with finite dissipation exists instead of the superlubric phase. An Aubry-type transition with a symmetry breaking signature occurs, when the system changes from the smooth-sliding to stick-slip regime[12,13].

With the advent of atomic and friction force microscopes and microbalances it became possible to study individual sliding junctions at the atomistic level[14–17]. These techniques have identified many friction phenomena at the nanoscale, but many key aspects of friction dynamics are not yet well understood due to the lack of probes that characterize the contact surfaces directly and *in situ*[1].

Laser-cooled and trapped ions have been proposed to emulate nanocontacts and to provide insights into the dynamics of friction processes[18–20]. In this scenario, the FK model is emulated by a chain of ions trapped in the harmonic potential of an ion trap, which is overlapped with an optical standing wave mimicking the corrugation potential. Signatures of an Aubry-type transition, that is, fragmentation and symmetry breaking of the periodic configuration of the ion chain, have been predicted, when the optical lattice depth increases above a critical value[19]. Another signature of the Aubry transition is the existence of a soft mode, that is, a vibrational mode whose frequency approaches zero at the critical point and drives the transition from pinned to sliding motion[12]. Such behaviour is also predicted for finite chains of ions in an external optical corrugation potential[21].

Recently, Bylinskii *et al.*[22] succeeded in cooling up to five ions into an optical lattice and demonstrated the onset of reduced friction and dissipation in a coupled atomic many-body system. In this experiment, the symmetry breaking Aubry-type transition has been observed for the first time with microscopic resolution[23], together with velocity effects in the stick-slip motion[24]. Another synthetic system, in which the microscopic processes of friction have become accessible, are colloidal monolayers driven across external optical potentials[25]. All these systems aim to emulate the classical FK model, where a layer of interacting particles slides over a fixed rigid corrugation potential.

Here we report on the microscopic and spectroscopic control of a system without an externally imposed corrugation potential but consisting of two deformable back acting atomic layers, whose relative motion exhibits the phenomena of nanoscale friction. This system has similarities to a refined microscopic model of friction, which replaces the rigid substrate by a deformable substrate monolayer pinned to a solid body[26]. In particular, we investigate static friction under the influence of a structural defect and demonstrate physical properties of the system, which are common to finite incommensurate systems. We use a structural defect (kink) in an ion Coulomb crystal[27] to create a local disturbance in the ion spacing in the upper and lower chain, and demonstrate an Aubry-type transition when the interatomic spacing of the layers is varied. We show, using numerical calculations, that the soft mode frequency exhibits a power law scaling behaviour in the vicinity of the critical point, where the system becomes superlubric. The experimental spectroscopic measurements show a small reduction in the frequency of the soft mode. The non-vanishing frequency of the sliding mode is due to the finite temperature exciting nonlinear dynamics. In addition, the experimentally measured order parameter agrees with numerical results, which also exhibit a power law scaling in the vicinity of the critical point. Our system relates to solid-state phenomena such as charge density waves[28] and dislocations in crystals[6]. In particular, the scenario of two interacting, deformable atomic chains with back action is analogous to friction in fibrous composite materials[29], sliding of DNA strands[30] and propagation of protein loops[31]. The manuscript is structured as follows: in the first section we introduce our experiment and model system. In the next section, we investigate the structural features of an Aubry-type transition—the symmetry breaking, the order parameter and the hull function. We then study the properties of the soft mode using spectroscopic measurements and numerical calculations. Finally, we discuss our results and prospects of our model system.

## Results

**Experimental system.** Our system consists of a two-dimensional ion Coulomb crystal in a linear rf trap[32,33]. Several tens of $^{172}Yb^+$ ions are laser-cooled to temperatures around $T \approx 1$ mK, so that they crystallize and self-organize into two ordered chains with interatomic distances of ca. 15 to 20 μm, see Methods. The ions are fluoresced by near-resonant laser light and imaged onto an electron multiplying charged coupled device (CCD) camera providing single atom and photon detection, as graphically depicted in Fig. 1a. The axial harmonic confinement of the ion trap leads to an inhomogeneous ion spacing, in particular at the edges of the crystal, while the central part of large-enough crystals exhibits a slowly varying lattice spacing with interatomic distances $a$ and $b$ within one layer and in between the layers, respectively, see Fig. 1b.

In the following, we consider the one-dimensional axial motion of the two chains along the $z$ axis in opposite directions. The friction dynamics depends on the relative interaction energies within each crystal row, $U_{intra}$, and between the rows, $U_{inter}$. To estimate the energy scales of these competing interaction strengths in our system, we relate them to the characteristic length and frequency scales in the Coulomb crystal. For this, it is convenient to view the system as two linear chains of identical particles of mass $m$ which are joined by interatomic springs of stiffness $\kappa$, depending on the inter-ion distance $a$. The masses are attached to a rigid support by external springs of stiffness $D = m \omega_{ax}^2$ due to the elastic confinement in the ion trap along the

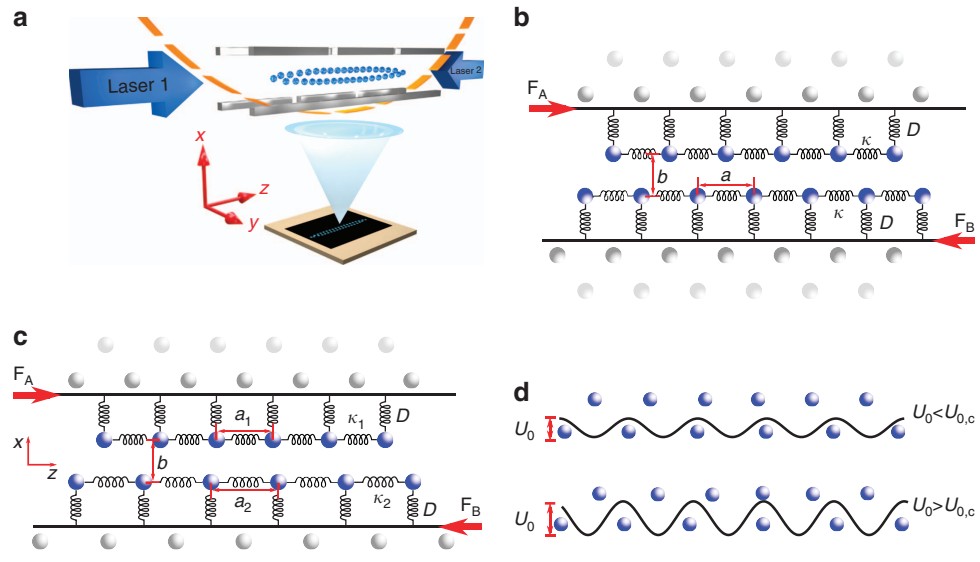

**Figure 1 | Experimental set-up and model system.** (**a**) Schematic of the experimental set-up. Several tens of ions are trapped inside a linear rf trap. An rf amplitude of around 1,000 V at the four quadrupole electrodes creates a time averaged confining potential in radial direction. The harmonic axial confinement is created via different dc voltages on the segmented electrodes, indicated by the orange dashed line. Typical radial and axial trapping frequencies are $\omega_{\text{rad}} \approx 2\pi\,140\,\text{kHz}$ and $\omega_{\text{ax}} \approx 2\pi\,25\,\text{kHz}$, leading to inter-ion distances of $a \approx 20\,\mu\text{m}$ and $b \approx 15\,\mu\text{m}$. The ions are illuminated with laser 1. Vibrational mode spectroscopy is performed with laser 2. The fluorescence of the ions is imaged onto an electron multiplying CCD camera. (**b**) The central part of the Coulomb crystal can be pictured as a self-organized interface between two solids. For the axial movement of the two atomic layers we consider the following model. Each chain is connected to the substrate via springs. The next-neighbour interaction between particles in the same row is modelled as spring forces. The deformability leads to identical intra chain spacing $a$ in upper and low layer. The chains can move relative against each other, using differential laser light forces. (**c**) A structural defect inside the Coulomb crystal locally breaks the periodicity of the two chains, resulting in different particle distances $a_1$ and $a_2$ in the chains. (**d**) The Coulomb potential of one row of ions acts as the corrugation potential for the other row. The depth of the corrugation $U_0$ determines the dynamics of the system. Below a critical corrugation depth $U_0 < U_{0,\text{c}}$ the system displays horizontal mirror symmetry. Above the critical value $U_0 > U_{0,\text{c}}$ the symmetry is broken.

axial direction. Here $\omega_{\text{ax}}/2\pi$ is the frequency of the centre of mass mode along the $z$ axis. In this setting, the two chains of ions can be moved and distorted by optical forces or light pressure as indicated with $\mathbf{F}_A$ and $\mathbf{F}_B$ in Fig. 1b,c.

For the case of periodically ordered chains as shown in Fig. 1b, stick-slip motion of the two layers of ions with respect to each other is expected to be approximated by the classical Prandtl–Tomlinson model[34], where a single particle has to overcome the potential energy barriers created by the periodic atomic lattice below. However, when a structural defect is present, the local disturbance in the periodicity of upper and lower chain leads to the nonlinear many-body phenomenon of reduced friction[6,7]. The underlying reason is that atoms in one layer will now locally sense different corrugation potential energies, which then can be stored by the internal springs described by $\kappa$. To investigate friction in soft chains of atomically flat layers under the presence of a defect, we create a stable structural defect in the centre of the Coulomb crystal, see Methods. This causes slightly different interatomic distances $a_1$ and $a_2$ in the central part of the Coulomb crystal, see Fig. 1c.

A rough estimate of the interaction energy between two ions inside one layer is $U_{\text{intra}} = \frac{1}{2}\kappa z^2$. The corrugation potential pinning ions of one chain with respect to the other chain can be locally approximated as $U_{\text{inter}} = \frac{1}{2} U_0 \left(\cos\left[\frac{2\pi}{a} z\right] + 1\right)$, as indicated in Fig. 1d. Depending on whether the interaction inside an atomic layer or between the layers is larger, a transition from the sliding to the pinned regime at a critical depth of the corrugation potential $U_{0,\text{c}}$ is expected. The interaction dynamics between the two atomic layers is thus governed by two competing energies. In harmonic approximation, they can be expressed in terms of frequencies $\omega_{\text{pinning}} = \sqrt{\frac{U_0}{m}\frac{2\pi^2}{a^2}}$ and $\omega_{\text{natural}} = \sqrt{\frac{\kappa}{m}}$, defining the corrugation parameter $\eta = \frac{\omega_{\text{pinning}}^2}{\omega_{\text{natural}}^2}$.

To estimate the ratio of the competing energy scales, we have numerically calculated $\omega_{\text{pinning}}$ and $\omega_{\text{natural}}$ for the central ions in the zigzag configuration, see Supplementary Note 1 and Supplementary Fig. 1. While $\omega_{\text{pinning}}$ depends on both interatomic distances $a$ and $b$, the natural frequency of an ion inside a layer can be expressed as $\omega_{\text{natural}} \sim \sqrt{\frac{e^2}{\pi\epsilon_0}\frac{1}{ma^3}}$, when considering only next-neighbour interaction. As we vary the relative interatomic distances $a$ and $b$, and with this the strength of the interaction potentials, we expect to observe a pinned to sliding transition. Experimentally this can be achieved by varying the ratio $\alpha = \omega_{\text{rad}}/\omega_{\text{ax}}$ of radial and axial trapping frequencies, that is, the aspect ratio of radial and axial confinement in the ion trap. $\alpha$ is used as the control parameter in our experiment to cross the transition at the critical point $\alpha_{\text{c}}$. For the numerical calculations we also plot $\eta$, which scales linearly with $\alpha$ close to the transition, see Supplementary Fig. 2.

**Symmetry breaking transition.** A general signature of Aubry and Aubry-type transitions, in both infinite and finite systems, is the breaking of analyticity[7]. This refers to the description of the sliding system in terms of a hull function, which parameterizes all possible configurations of the ground state under the presence of a sliding force. This function becomes non-analytic when the Aubry transition is crossed. In our system, as the two ion chains are brought closer together the corrugation depth increases and

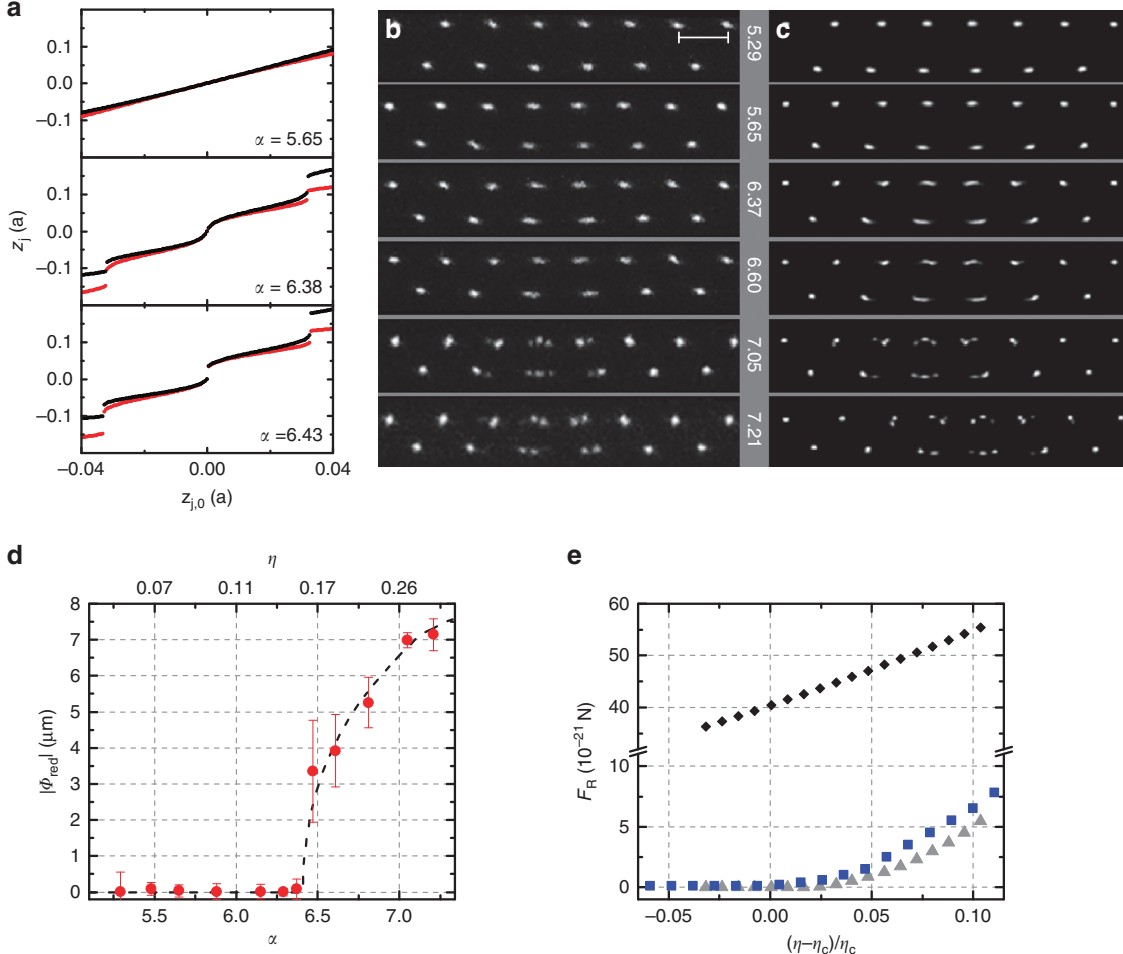

**Figure 2 | Symmetry breaking of the crystal and restoring force. (a)** Numerically calculated hull functions in units of the lattice constant $a$ for the two central ions, left (red points) and right (black points) to the crystal centre. Below $\alpha_c = 6.41$ the hull functions are continuous. After the pinning transition is crossed they exhibit a central gap. Slightly below $\alpha_c$ secondary gaps are observed. The inhomogeneity of the crystal leads to a lower charge density, and thus lower $\omega_{natural}$, further away from the crystal centre. If the defect is not at the centre, the pinning transition occurs for $\alpha < \alpha_c$. The secondary gaps of the central ions are the response to the analyticity breaking of the outer ions' hull functions. **(b)** Photos of experimentally observed 30-ion crystal configurations at different aspect ratios $\alpha = \frac{\omega_{rad}}{\omega_{ax}}$ from 5.29 to 7.21, as indicated in the grey bar. Relative errors are less than 1%. The exposure time is 700 ms. Laser 1 continuously cools the ions. No force is applied. The blurring of the ion positions near the sliding transition and the appearance of multiple configurations above it are due to thermal excitations. The scale bar is 16.5 μm. **(c)** Numerically simulated 30-ion crystal configurations at $T = 1$ mK integrated over 10 ms of time evolution. **(d)** The absolute value of the order parameter $|\Phi_{red}|$ for 30 ions plotted against $\alpha$ and $\eta$. Experimental data (red circles) are shown in comparison to numerically obtained values for $T = 0$ K (black dashed line). Experimental values represent a weighted average over 5–26 measurements, with exception of $\alpha = 7.21$, where only 2 configurations were observed. Error bars are one standard deviation weighted by fit errors. **(e)** Numerically calculated restoring force $F_R$ plotted against $(\eta - \eta_c)/\eta_c$ for an inhomogeneous crystal (blue squares), a homogeneous crystal (grey triangles) and an ideal zigzag without defect (black diamonds). $\eta$ refers to the inhomogeneous case. The parameters for the homogeneous and commensurate crystals were chosen to have an identical ratio of inner inter-ion distances $a,b$. The friction force needed in the ideal crystal slightly above the pinning transition is roughly an order of magnitude bigger.

eventually reaches a critical point. Above this point the corrugation prevents the ions from assuming all positions during sliding and their trajectories, and thus the hull function, become discontinuous. This is the point where Peierls–Nabarro (PN) barriers are formed[6].

To study the analyticity breaking in our system, we first conduct numerical simulations of a 30-ion crystal under the presence of differential forces applied to upper and lower ion chain. In the classical FK model, the hull function $z_j(z_{j,0})$ is defined as the coordinate of a particle $j$ under the influence of a static corrugation potential in relation to its unperturbed coordinate $z_{j,0}$ without the underlying lattice. For the self-organized system of two interacting atomic chains, the corrugation potential is given by the Coulomb potential of

the 2nd row of ions and thus cannot be switched off. To calculate the hull function in such a system, we first apply opposite forces $\pm \mathbf{F}/2$ to each row of the crystal to obtain $z_j$. Then we apply the force $\mathbf{F}$ to both rows in the same direction, moving the lattice along with the ion to obtain $z_{j,0}$, that is, the equilibrium position of the ions in the harmonic trapping potential without any influence of the underlying lattice. We implement this principle in our numerical simulations and show the result in Fig. 2a. A more detailed evolution of the hull function can be found in the Supplementary Movie 1. For the same control parameter $\alpha$, where a primary gap opens up in the hull function at $z_{j,0} = 0$, we observe a structural symmetry break in the equilibrium positions of the crystal configuration. This is visible in Fig. 2b, which shows photos of experimental realizations of 30-ion crystals at

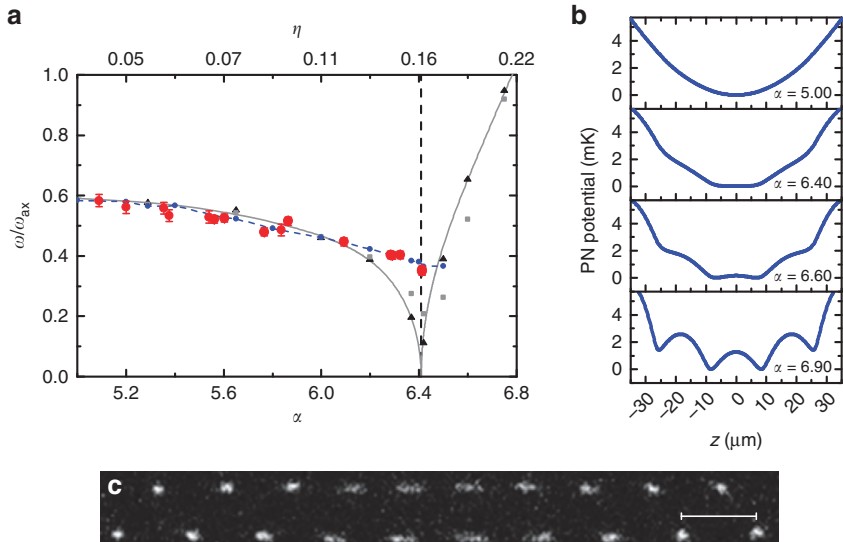

**Figure 3 | Vibrational soft mode and PN barriers. (a)** Frequency of the soft mode in a 30-ion crystal. The experimental data is shown in red circles and the error bars are given by uncertainties of the measured soft mode and common mode frequencies. The solid line displays the numerically calculated dispersion relation at $T = 0$ K. Frequencies extracted via a Fourier transformation from molecular dynamics simulation are given by black triangles at $T = 5$ μK, grey squares at $T = 50$ μK and blue circles at $T = 1$ mK. The dashed blue line acts as a guide for the eye. All frequencies are plotted in units of the axial secular frequency $\omega_{ax} = 2\pi(25.6 \pm 0.2)$ kHz. The pinning transition is marked with a vertical line. **(b)** Numerically calculated PN potentials showing the onset of barriers above $\alpha_c$. At temperatures of $T = 1$ mK the thermal fluctuations sample the multiple minima of the PN potential and consequently no harmonic motion with a single distinct frequency can exist for $\alpha > \alpha_c$. **(c)** A CCD image of an ion Coulomb crystal in our experiment in which the localized vibrational mode is resonantly excited by a focused laser beam. It is taken at $\alpha = 5.77$ with an exposure time of 100 ms and 300 μW of power in the cooling laser. The focused laser beam is modulated between 0 and 35 μW with a frequency of $12.1 \pm 0.3$ kHz. Scale bar, 20.5 μm.

different $\alpha$. They are compared to images obtained from numerical simulations at $T = 1$ mK, shown in Fig. 2c. In both the experiment and the simulations no driving force is applied to the crystals. The motional excitation of the ions is due to the finite temperature.

The collective dynamics in a discrete nonlinear system is typically described by the PN potential[6]. It corresponds to the energy needed to move the charge density. For temperatures close to $T = 0$ K the system would choose one stable configuration in one of the minima of the PN potential, shown in Fig. 3b. For finite temperatures close to the height of the PN barriers, the different choices of the system become visible. In Fig. 2b,c for $\alpha < \alpha_c$ the crystal shows horizontal mirror symmetry around its centre. As $\alpha$ approaches $\alpha_c$ the thermal oscillation amplitude of the inner ions becomes larger, indicating that the ions are less pinned close to criticality. At $\alpha_c = 6.41$ the pinning transition is crossed. At this point the PN potential develops multiple minima, as can be seen in Fig. 3b at $\alpha > 6.4$, causing the symmetry breaking of the crystal. A high-resolution video of the emergence of PN barriers is provided in the Supplementary Movie 2. Slightly above $\alpha_c$ we do not resolve the symmetry broken configuration, as thermal fluctuations mask the local minima. However, a larger spread in the central ion positions is observed. As $\alpha$ is increased over 7.00, multiple stable crystal configurations become visible, indicating the existence of multiple minima in the PN potentials.

From the crystal configurations we can extract a structural order parameter that quantifies the symmetry breaking at the transition from pinned to sliding. For our scenario, that is, the sliding of two deformable chains, we choose to define an order parameter $\Phi$ as the relative axial distance between ions of different layers, in the following labelled Chain A and B. It characterizes the horizontal mirror symmetry of the system,

$$\Phi = \sum_{i \in \text{Chain A}} \text{sgn}(z_i) \cdot \min_{j \in \text{Chain B}} |z_i - z_j|, \quad (1)$$

where $z_i$ is the axial coordinate of the ion i and $z = 0$ is the axial symmetry axis below the transition point. In the sliding regime $\Phi = 0$. Crystal configurations obtained from numerical simulations at $T = 0$ K, show that at the critical point around $\alpha_c \approx 6.41$, the order parameter shows a cusp where the system chooses a state of broken symmetry, a typical signature of a second-order phase transition, see dashed line in Fig. 2d. When evaluating the experimental data, to reduce the accumulated error of fitted ion positions and to avoid errors due to aberrations of our imaging system at large distances from the optical axis, we measure and plot a reduced version $\Phi_{\text{red}} = \Phi_{i=N/2} + \Phi_{i=N/2+1}$, which includes only the central terms with the largest contribution to the overall sum. The experimental data in Fig. 2d (red circles) are extracted from CCD images, some of which are shown in Fig. 2b. The experimentally and numerically observed critical corrugation parameter $\eta_c \approx 0.16$ is smaller than 1, because for simplicity the interaction energies were calculated only for the zigzag. The difference between the interatomic distances $a$ and $a_{1,2}$ explain the smaller critical corrugation parameter.

At last, following Benassi et al.[19] and using the data from numerical simulations, we calculate the restoring force $F_R$, which is needed to restore symmetry in our system, above the transition point. $F_R$ can be identified with the static friction force $F_S$ in an infinite and thus homogeneous chain. $F_S$ is needed to overcome the PN barriers and translate the localized defect by one lattice site. In Fig. 2e, $F_R$ is shown for a homogeneous and inhomogeneous crystal with defect, as well as an ideal zigzag. The restoring forces are similar, indicating that our model system, consisting of finite and inhomogeneous Coulomb crystals, is a good approximation to large scale systems, with homogeneous particle spacing. Comparing the axial inter-ion distances near the defect, we find that distances between the innermost ions differ by less than 1%. For next-neighbour ions the difference increases to a few per cent. Compared to the force $F_S \approx 5 \times 10^{-20}$ N needed to move two perfectly matched chains (a zigzag configuration

without defect) against each other, the static friction force is reduced by more than an order of magnitude slightly above the pinning transition. The derivation of $F_S$ for the scenario of unperturbed crystals, analogous to the Prandtl–Tomlinson model, is detailed in the Supplementary Note 2.

**Soft mode and critical scaling.** The sliding of two atomic chains is driven by the vibrational axial shear mode of the crystal, where the two rows move in opposite directions. Its frequency being zero signifies it costs no energy to translate the layers relative to one another. For finite systems, the frequency of the lowest vibrational mode approaches zero but remains finite both above and below the pinning transition and thus the system is only superlubric at the critical point[12]. In an ideal zigzag however no such soft mode exists, as the system is commensurate. Only when a structural defect is present in the lattice disturbing the regular ordering of particles a soft mode is present. It is localized at the position of the defect[35] and kink dynamics governs the sliding of the two atomic layers.

We first calculate the dispersion relation of the vibrational modes in our two-dimensional ion Coulomb crystal for different interatomic distances $b$ between the layers, see Methods. The solid grey line in Fig. 3a shows the dependence of the localized soft mode on the control parameter $\alpha$ and the corrugation parameter $\eta$. For $\alpha < 6.8$ this mode is the lowest frequency mode in the crystal. Its frequency reaches zero at the transition point, indicated by a vertical dashed line, and assumes finite values in the sliding regime. In the experiment we use differential laser light forces to resonantly excite the vibrational modes of the ion Coulomb crystal by sinusoidal intensity modulation. An experimental photo in which the soft mode is excited is shown in Fig. 3c. Further details are found in the Methods. The measurement results are depicted as red circles in Fig. 3a. Below the sliding-to-pinning transition, our measurements agree with the calculated frequencies of the dispersion relation. Close to the transition the experimental measurements deviate from theory. Above the critical point no excitation of the soft mode was possible. To understand this behaviour, we conducted molecular dynamics simulations of the unperturbed crystal, that is without laser excitation, at finite temperatures (for details see Methods). From the ion trajectories the vibrational spectrum of the crystal was extracted using a Fourier transformation. To benchmark our analysis, we first perform simulations at $T = 5\,\mu K$ and find the results to be in good agreement with our calculations. At an increased temperature of $T = 50\,\mu K$ the Fourier spectrum of the ion vibrations deviates from the dispersion relation. For $T = 1\,mK$ and $\alpha > \alpha_c$ we observe a broad range of frequencies in the Fourier

transform with no clear resonance. For $\alpha < \alpha_c$ the soft mode frequencies extracted from these simulations agree with the experimentally observed resonances. We attribute the deviation of the observed frequencies near the transition point to the increasing contribution of the nonlinearities of the PN potential, which is shown in Fig. 3b. Above the symmetry breaking transition the thermal amplitude of the kink motion overcomes the barriers between potential minima, and no single distinct mode frequency exists. For consistency, we compare the experimentally observed thermal amplitudes of the central ions to numerical simulations similar to Fig. 2b. From this we obtain an estimate for the temperature of the crystal, which is found to be $T = 0.5 \pm 0.4\,mK$.

The finite size of ion Coulomb crystals in a harmonic trap and thus the inhomogeneous charge density results in a global curvature of the PN potential, as can be seen in Fig. 3b. This is in stark contrast to the vanishing PN potential for infinite systems below criticality. Not only is the crystal finite, but we expect the sliding dynamics to be governed by the local distortion of the structural defect. To discern whether critical scaling exists in this system, we numerically calculate the soft mode frequency and the order parameter in the vicinity of the sliding-to-pinning transition with high resolution. Results for different crystal sizes are shown in Fig. 4. We fit the order parameter as $\Phi \propto (\eta - \eta_c)^\sigma$ and the soft mode frequencies above and below critical point as $\omega^\pm \propto (\eta - \eta_c)^{\chi^\pm}$. We find that independent of the ion number and the crystal size $\sigma \approx \chi^\pm \approx 0.5$, similar to what has been observed for an Aubry-type transition in finite systems[12]. In the Supplementary Fig. 3 we show a linear presentation of the soft mode for 30 and 60 ions.

## Discussion

In this work, we use an ion Coulomb crystal with a structural defect to experimentally and numerically study static friction in a self-organized system. In the scenario emulated by our model system two deformable chains slide on top of each other. Such situations frequently arise in nature, in particular in biomolecules[30,31]. We experimentally observe the symmetry breaking at the sliding-to-pinning transition and we spectroscopically resolve the frequency of the localized vibrational mode, which is responsible for charge transport in our system. The strength of trapped and laser-cooled ions is that they are a readily accessible model system, in which many-body physics can be observed with single atom resolution. With high quality imaging optics, a spatial resolution of a few nm can be achieved[36]. Single atom resolution also allows studies of kinetic friction[20,37] and the spectroscopic access to internal degrees of freedom enables

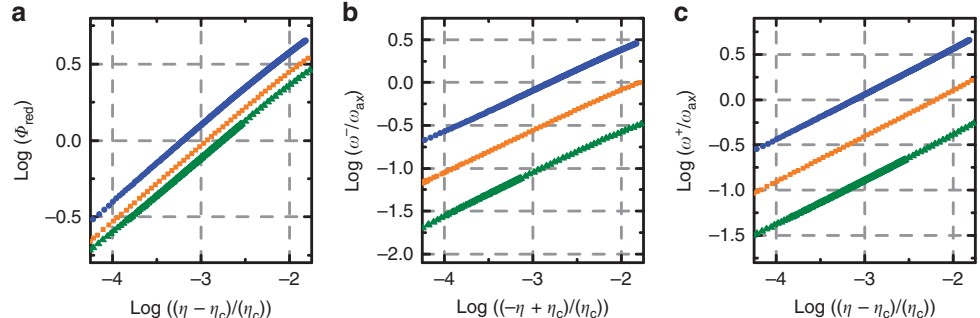

**Figure 4 | Critical scaling.** All graphs show numerical results. (**a**) Double logarithmic plot of the order parameter for $N = 30$ (blue points), 60 (orange points) and 90 (green points) ions. The fitted critical exponents are $\sigma_{30} = 0.50$, $\sigma_{60} = 0.50$ and $\sigma_{90} = 0.49$. (**b**) Double logarithmic plot of the soft mode frequency for $\eta < \eta_c$. The colour code is identical. The plots for $N = 30$ and 60 are shifted by 1 and 1/2 for clarity. Fitted critical exponents are $\chi^-_{30} = 0.486$, $\chi^-_{60} = 0.491$ and $\chi^-_{90} = 0.504$. (**c**) Double logarithmic plot of the soft mode frequency for $\eta > \eta_c$. The same offsets apply. The fitted critical exponents are $\chi^+_{30} = 0.498$, $\chi^+_{60} = 0.501$ and $\chi^+_{90} = 0.490$.

investigations of non-equilibrium dynamics and transport processes with ns and μs time resolution[38]. Typically, trapped ions are confined in harmonic potentials which lead to an inhomogeneous ion spacing. Our results show that the inhomogeneity due to the boundary conditions and the system size influence our model system only slightly. This is due to the fact, that we focus on the phenomena of the onset of sliding, which is dictated by the localized lattice defect. However, given the controllability of Coulomb crystals it is possible to extend our model to versatile geometries, e.g., equally spaced crystals in anharmonic potentials or ring traps[39]. Moreover, the ideas presented in this work may be used to investigate the currently poorly understood Aubry transition in two-dimensional systems[40], since ion traps can trap large three-dimensional Coulomb crystals composed of regular two-dimensional layers[41]. Furthermore, by investigating the soft mode of the sticking to sliding transition, we demonstrate the spectroscopic observation of the highly nonlinear vibrational mode of the structural defect. These localized modes have also been proposed for the implementation of quantum information protocols[35]. Recently, the high energy gap mode and its coupling to the low frequency mode of the localized defect were observed (T. Schaetz, personal communication).

In the future, the experiments can be improved by further cooling to the μK regime using narrow transitions[32] or dark resonances[42,43], making quantum effects of friction accessible. This can provide a new platform for studying the physics of Wigner crystals[18,44].

## Methods

**Coulomb crystals and structural defect creation.** Coulomb crystals form, when ions are cooled to kinetic energies lower than the potential energy of the Coulomb system. This is achieved by laser cooling the $^{172}$Yb$^+$ ions on the broad, dipole allowed atomic transition $^2$S$_{1/2}$ to $^2$P$_{1/2}$ at 370 nm with a natural linewidth of $\gamma = 20$ MHz. The frequency of the cooling laser is detuned from resonance by $\delta = -\gamma/2$ resulting in a crystal temperature close to the Doppler cooling limit of $T = 0.5$ mK. The cooling laser illuminates the ions with a power between 200 and 300 μW. The waist is 2.56 mm in axial direction and 80 μm in radial direction. One to three-dimensional crystal configurations can be chosen, depending on the relative strength of the harmonic trapping potentials $U(r) = \frac{1}{2}\omega_{rad}^2 m r^2$ in radial direction and $U(z) = \frac{1}{2}\omega_{ax}^2 m z^2$ in axial direction[45,46], where $m$ is the ion mass and $r = \sqrt{x^2 + y^2}$. When the structural transition from the linear chain to the two-dimensional zigzag is crossed non-adiabatically, defects can be created[27,47–49]. In our experiment, we create and stabilize a structural defect in the centre of the Coulomb crystal by fast ramps of the radially confining rf potential[27]. Typical radial and axial trapping frequencies are $\omega_{rad,x} \approx 2\pi\,140$ kHz and $\omega_{ax} \approx 2\pi\,25$ kHz, leading to inter-ion distances of $a \approx 20$ μm and $b \approx 15$ μm. The anisotropy of the radial confinement with $\frac{\omega_{rad,y}}{\omega_{rad,x}} \approx 1.3$ creates a two-dimensional setting for the frictional dynamics. The trap frequencies are typically determined within 100 Hz, amounting to an error of $\pm 0.02$ in the control parameter $\alpha = \frac{\omega_{rad}}{\omega_{ax}}$. The ions are imaged onto an electron multiplying CCD camera using a self-built detection lens with an $NA = 0.2$ and a magnification of 24. In a regular zigzag configuration, we can resolve the ion positions within 40 nm at exposure times of 700 ms by fitting a Gaussian profile to the images. The resolution is limited by the magnification of our imaging system and the pixel size of the CCD chip. Fitting multiple ion positions in the symmetry broken regime runs into a limit close to the transition where the intensity maxima are separated by less than a pixel.

**Spectroscopy.** For the spectroscopy of the vibrational modes, we use a second laser beam under an angle of 25° to the crystal axis, which is focused to a waist of $ca.$ 80 μm. The laser is amplitude modulated with a frequency $\nu$, exerting a periodically oscillating force $F = F_0 \cos[2\pi \cdot \nu t]$ onto the ions. To efficiently excite the shear mode, we centre the laser beam axially and slightly misalign it along the radial direction to obtain a differential light force in axial direction between the two chains. If $\nu$ is near-resonant with a vibrational mode, a broadening of the ion positions is observed. To determine the resonance, we sweep $\nu$ with a speed between 1 and 2 kHz s$^{-1}$. The full-width at half maximum of the resonances is about 1 kHz and its centre frequency is determined within 300–400 Hz.

**Numerical simulations.** For simulations with $T = 0$ K, we determine the dispersion relation by diagonalising the Hessian matrix $H_{ij} = \frac{\partial^2 V}{\partial q_i \partial q_j}\big|_{\mathbf{q}(0)}$, where $V$ is the

potential energy; $q_i$ are the degrees of freedom with i ranging from 1 to $2N$, with $N$ being the number of ions; $\mathbf{q}(0)$ represents the equilibrium configuration. The eigenvectors $H_{ij}$ are the vibrational normal modes and the corresponding eigenvalues are squares of the normal frequencies. The equilibrium positions $\mathbf{q}(0)$ are found by solving the equations of motions numerically using gradient descent methods. For the calculation of the hull function and the restoring force $F_R$ we use the same method, but add axial differential forces between the two chains to the equations of motion.

For simulations at non-zero temperature we are solving the Langevin equation, which includes the harmonic motion of the ions in a ponderomotive trapping potential under the presence of a stochastic force $\varepsilon(t)$. The fluctuation-dissipation relation $\langle \varepsilon_{zj}(t)\varepsilon_{\beta i}(t')\rangle = 2\eta k_B T \delta_{\alpha\beta}\delta_{ij}\delta(t - t')$, with $\alpha$, $\beta = x$, $y$, $z$ and $\eta$ the friction coefficient, connects the stochastic force to the temperature $T$ of the system[27].

The PN potential is calculated by finding the adiabatic trajectory of the defect and extracting the potential energy. Finding the trajectory is an optimization problem, which is solved using the method of Lagrange multipliers[50].

**Data availability.** The data that support the findings of this study are available from the corresponding author upon reasonable request.

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

## Acknowledgements
We thank Jonas Keller for valuable help with the experiment control and many useful discussions, and Lars Timm for supporting numerical simulations. We thank Erio Tosatti and Andrey Surzhykov for reading of the manuscript. This work was supported by EPSRC National Quantum Technology Hub in Networked Quantum Information Processing (R.N.) and by DFG through grant ME 3648/1-1.

## Author contributions
The experiment was initiated and led by T.E.M. R.N. developed the numerical codes. R.N., J.K. and T.S. carried out the simulations. D.K. and J.K. designed the experiment with input from T.E.M. D.K. and J.K. carried out the experiments and performed the data analysis. All authors contributed to the discussion of results and participated in the manuscript preparation.

## Additional information

**Competing interests:** The authors declare no competing financial interests.

**Publisher's note**: 

