## [Peer Review File · Nature Communications]

Reviewers' comments:

Reviewer #1 (Remarks to the Author):

The manuscript "Probing Nanofriction and Aubry-type signatures in a self-assembled system" describes measurements and modeling of a Coulomb crystal of trapped ions. The authors adjust the trapping potential such that in the center of the ion crystal two strings of ions form. The commensurability of the two crystals is broken by dislocation created by a diabatic transition from the linear to the zigzag phase. The experimental results are impressive and are reproduced well with numerical models.

The manuscript is well written and discusses much of the related physics in reasonable depth. The trapped ion system is a promising route to study this interesting system. I thus find the manuscript of high interest and because of the importance and interdisciplinary nature of the phenomenon of friction suitable for publication in nature communications.

However, I think some improvements could and should be made first.

Generally: While I appreciate that the experimental observations are complemented with detailed numerics, it is crucial to make clear what is experiment and what is numerics. At the very least this should be stated in the figure caption, see Figs 2c, 2e?, 4a.

line 68/69: the authors claim to report on two atomic layers sliding on top of each other. I do not see evidence for the actual sliding. Yes, the physics the authors study is about the onset of this sliding, but the sliding (over a substantial distance) itself is in my opinion not reported. For this it would be necessary that the whole layer would move by one lattice site.

line 76: how can a soft mode have a power law scaling? It is not understandable at this point. I guess the authors mean the frequency scales? Further, there is only numerical evidence for this (if I understand correctly), so it should be made clear that those are not measurements.

line 78: The observed frequency of the soft mode is not only explained by temperature. I think this statement is not understandable at this point either. It may be best to either leave out this statement or say that a finite temperature effects are observed.

line 85 and following: it is not clear to me how relevant the inhomogeneity is in all these experiments. I think this should be addressed in detail.

line 95: "Its frequency depends on .." The previous sentence just stated that its frequency is zero. This bears some explanation for instance that it is not 0 in a finite size system, etc.

Line 173: I do not understand the sense in the procedure "Then we apply both forces in the same direction ..." This makes no sense to me as the relative positions of the two layers will not change and hence this force should not do anything other than to accelerate both crystals together.

line 183:

fig 2a: what do the uncertainties mean? What is a ? The lattice spacing? Should it have units?

fig 2c: why is the hull function only displayed from -0.04 till 0.04, should it not go from -0.5 to 0.5? I guess there is nothing interesting happening there. But if nothing interesting is happening there, then I start to wonder whether the present system is not in some weird limit of the Frenkel-Kontorova model. Finally, is this a measurement?

fig 2e: I am guessing that those come from simulations. This should be made clear right from the beginning. It left me pondering how the hell this was measured for quite some time. The implicit statement the following sentence is not sufficient.

fig 4: indicate that those are numerical results.

Reviewer #2 (Remarks to the Author):

This combined experimental and numerical study reports on the frictional properties of two interacting and incommensurate layers of an ion Coulomb crystal. In the case considered here, the incommensurability is introduced by a single topological defect, i.e. a kink. When the electrostatic coupling between the layers is gradually decreased, a soft vibrational mode is observed which corresponds the disappearance of static friction in agreement with an Aubry transition. Both, the soft mode and the structural order parameter display a power-law close to the critical point. These findings are corroborated by numerical simulations.

The Aubry transition is an intriguing phenomenon which has been rigorously, i.e. mathematically proven for infinite, one-dimensional systems. When reading the paper, I first thought, that the authors have performed measurements on a two-dimensional system (because they repeatedly refer to layers), however, then I realized, that they only considered the case of two chains of interacting particles. I understand, that their work adds additional information beyond a recent similar study of Bylinskii et al., but I do not consider the results of this manuscript not as a substantial improvement compared to the other one. Therefore, I do not recommend it for Nat. Comm. but suggest to submit this to another, more specialized journal.

Below, please find my questions and remarks:

1. As the authors certainly know, the nature of the Aubry transition changes from continuous two first-order when going from 1D to 2D. When looking at their experimental situation (Fig.1a), my feeling is, that they are somewhere in between. What other evidence the authors have, that a 1D description is valid?
2. Compared to previous work where the Aubry transition was demonstrated in a system of 5 ions only, here the number of ions is somewhat larger but it is still far away from the infinite case considered by Aubry. How does the finiteness of the system affect the way how the soft mode develops? How Fig.3a would vary if one, lets say doubles the chain length?
3. The authors say, that the axial harmonic confinement of the ion trap leads to an inhomogeneous particle spacing but that the central part of the crystals exhibits a (quasi)-periodic lattice. What do the authors mean by the word "quasi"? I assume, that they do not refer to quasiperiodic order, but that the particle distance continuously changes. Then, however, I do not really understand the meaning of incommensurability.
4. The presence of a soft mode is indeed a signature of the Aubry transition. In contrast to Aubry, who considered an elastically coupled chain of particles in a strictly periodic substrate potential, here the authors consider the situation of two interacting chains of particles. Similar to my question 1, how such soft modes are influenced by considering also further vertical couplings between the particles?

Reviewer #3 (Remarks to the Author):

The manuscript reports a study of stick-slip friction using a two-layered ion crystal in a zig-zag

arrangement. The idea to translate two ion chains against each other to implement the Frenkel-Kontorova model is ingenious. The paper reports interesting physics in a novel system, and sheds new light on microscopic friction models in finite systems. As such, the paper is clearly suited for Nature Communications.

However, I think that the presentation could be substantially improved. The current version of the manuscript is difficult to follow for even for the specialist, let alone for the general reader. The manuscript jumps back and forth between theoretical expectations, experimental results, and numerical modeling, and does not always make clear which is which. Furthermore, and perhaps relatedly, some of the main claims of the abstract and manuscript are misleading, and appear not to be substantiated by experimental results. For instance, the abstract states: "We show that local defects in the periodicity of the atomic layers cause a transition from sticking to sliding with Aubry type signatures." To the best of my understanding of the manuscript, the authors do not show a comparison of friction with and without local defects, as this sentence seems to imply, but only that there is a transition in the distribution of ion positions in the presence of a defect when the ratio of trap frequencies is varied. Perhaps the authors meant something along the lines: "We show that in a system with local defects resulting in incommensurate layers, there is a transition from sticking to sliding with Aubry type signatures."

More importantly, the abstract claims: "We demonstrate spectroscopic frequency measurements of the vibrational soft mode driving this transition and ... Both [one of them being the vibrational mode] exhibit critical scaling near the transition point." Now the experimental data in Fig. 3a only show a slow weak and linear reduction of the "soft-mode" frequency up to the critical point, with nothing particular happening at the presumed critical point. In numerical simulations also shown in this figure, this absence of an observation is explained by the finite, and relatively large, temperature of the system. Nonetheless, the claim of "critical scaling near the transition point" is clearly not borne out at all by the experimental data. (There is even no convincing mode softening at the critical point.) All claims of critical scaling derive purely from numerical simulations in Figures 3 and 4. So clearly the abstract and main text need to be reformulated to make clear what was, and what was not, observed. Critical scaling was not observed in the experiment, only in numerical simulations. The caption of Figure 4 should also state that this is a simulation.

Throughout the rest of the manuscript there are similar confusing jumps back and forth between experiment, numerical simulations, and theoretical expectations from the model. Many of the further reaching claims, such as the ones discussed above or the friction force in Fig. 2e, are only supported by numerical simulations, not by the experimental data.

The manuscript also does not explain how the experiment was actually performed. How and with what time dependence was the displacement force applied, or are the actual data averages over many different fixed forces? This should be stated in a main text. (p. 5, the caption to Figure 1 states "The chains can move against each other, e.g. using laser light forces." Why "e.g." here, what was actually done?) Was the displacement force for Figure 2 applied with a sinusoidal time dependence? If so, would the ions not spend more time at the turning points, and therefore a gap in the position distribution might appear that mimics the gap of the Aubry transition? Why do the ion positions not appear to be smeared out in Fig. 2a (I) if this represents the sliding phase? Is the amplitude of the sliding motion too small to see in the picture 2a(I), or was the amplitude changed between (I) and (VI)? Does Fig. 2C represent experimental data or merely numerical simulations? If the latter is the case, this should be pointed out in the figure caption.

Overall, the manuscript should be improved to make the experimental procedure and achievements clearer. My suggestion would be to first discuss the experimental results, and then separately their theoretical interpretation, so that it is clearer what has been experimentally achieved, and how. As far

as I can tell, the main experimental achievements are 1) the opening of the gap in position or order parameter as presented in Figure 2d, and the observation of a modest softening of the vibrational mode as presented by the red circles in Figure 3a. Since according to calculations one should observe a real dip in the transition frequency at the critical point, it should also be discussed what limits the temperature, and if there are prospects to repeat the measurement at lower temperature. The information presented in all other figures appears to be coming exclusively from numerical calculation, but it is not always clear, as in Figure 2c.

Overall, I quite like this work and the idea behind it, and I recommend publication in Nature Communications. However, I think that the authors could significantly strengthen the impact of their work by restructuring their manuscript to improve its readability, and by clearly stating claims that are substantiated by experimental data.

REVIEWERS' COMMENTS:

Reviewer #1 (Remarks to the Author):

All my concerns have been addressed satisfactory. I recommend publication in Nature Communications.

Reviewer #2 (Remarks to the Author):

When answering my questions regarding the dimensionality of system, the nature of the phase transition and possible finite size effects (questions 1- 3), the authors only argue with their simulations. I have to admit, that I was hoping for arguments which are based on their experimental rather than their numerical data. In that sense, I am not fully convinced in particular because the title of the paper (Probing nanofriction and Aubry-type signatures ...) clearly suggest experimental evidence for the Aubry transition.

Having read the comments of the other two referees, however, I do not want to play the devils advocat but recommend (similar to referee 1), that the authors clearly state what is experimental and what is numerical data.

With such changes, I am happy to recommend the manuscript for publication in Nat. Comm.

Reviewer #3 (Remarks to the Author):

The authors have responded adequately to the referees, and I find the manuscript substantially improved, and much clearer. There are only a few remaining small issues:

- i) Upon re-reading the title, "self-assembled system" seems a little strange, after all, the ions are trapped in an external trap, and the crystal does not "self-assemble", it at most "self-organizes" under the influence of laser cooling. However, I don't feel strongly about this issue.
- ii) In their response to me regarding what is shown in Figure 2, the authors reply:" Fig. 2a and b (now 2b and c) display pictures of the equilibrium positions of ion Coulomb crystals at different aspect ratios of the ion trap alpha averaged over 700 ms exposure time. No force is applied the motion here is purely temperature driven. At finite temperatures the ion positions are smeared out depending on how close the system is to the critical point. The gapped ion positions are explained by appearance of multiple stable crystal configurations after the crossing of the critical point. The thermal energy of the ions allows to overcome the PN barriers and thus makes the various configurations visible." This response is very clear, but the Figure caption of Figure 2 is not, i.e. it is not clear that it is the thermal energy rather than an applied force that makes the various configurations observable. I recommend that the authors modify the caption along the lines of the above response.
- iii) Line 194: "The experimentally and numerically observed critical corrugation parameter ...is smaller than 1, because for simplicity the interaction energies were calculated only for zigzag." I understand how this approximation can affect the calculation, but how can it affect the experimentally observed critical corrugation parameter?

This is an interesting and novel microscopic study of friction that is of interest to the general audience. In summary, I recommend publication of this manuscript in Nature Communications with minor revisions.

Reply to reviewer #1:

"The manuscript "Probing Nanofriction and Aubry-type signatures in a self-assembled system" describes measurements and modeling of a Coulomb crystal of trapped ions. The authors adjust the trapping potential such that in the center of the ion crystal two strings of ions form. The commensurability of the two crystals is broken by dislocation created by a diabatic transition from the linear to the zigzag phase. The experimental results are impressive and are reproduced well with numerical models.

The manuscript is well written and discusses much of the related physics in reasonable depth. The trapped ion system is a promising route to study this interesting system. I thus find the manuscript of high interest and because of the importance and interdisciplinary nature of the phenomenon of friction suitable for publication in nature communications.

However, I think some improvements could and should be made first."

The referee correctly summarises the contents of the paper. We are pleased that the referee finds the manuscript of high interest and relevance to the wider community, recommending its publication in Nature Communications. In the following we have addressed his or her comments and suggestions.

(Line numbers refer to the originally submitted manuscript)

Note: we have reordered Fig. 2a,b,c in the resubmitted version to be more coherent with the main text.

"Generally: While I appreciate that the experimental observations are complemented with detailed numerics, it is crucial to make clear what is experiment and what is numerics. At the very least this should be stated in the figure caption, see Figs 2c, 2e?, 4a."

We agree that the manuscript would benefit greatly from more clearly stating whether the results come from simulation or from experiment. We have now explicitly stated in the captions of all figures the nature of the results.

We have also modified several sentences in the main text to be clearer, e.g. line 176 reads now:

"This is visible in Fig. 2b, which shows photos of experimental realizations of 30 ion crystals at different α . They are compared to images obtained from numerical simulations at $T = 1$ mK, shown in Fig. 2c."

or when describing Fig. 2d it reads now "The experimental data in Fig. 2d (red circles) are extracted from CCD images, some of which are shown in 2b."

"line 68/69: the authors claim to report on two atomic layers sliding on top of each other. I do not see evidence for the actual sliding. Yes, the physics the authors study is about the onset of this sliding, but the sliding (over a substantial distance) itself is in my opinion not reported. For this it would be necessary that the whole layer would move by one lattice site."

It was certainly not our intention to suggest that our study reports the sliding over substantial distance. This sentence was merely used to introduce the idea of using a self-organized system for studying friction phenomena. The specifics of our findings are summarized later in the same paragraph.

We have changed the wording of this sentence so that it now reads: "Here, we report on the microscopic and spectroscopic control of a system without an externally imposed corrugation potential but consisting of two deformable back acting atomic layers, whose relative motion exhibits the phenomena of nanoscale friction."

"line 76: how can a soft mode have a power law scaling? It is not understandable at this point. I guess the

authors mean the frequency scales? Further, there is only numerical evidence for this (if I understand correctly), so it should be made clear that those are not measurements."

and

"line 78: The observed frequency of the soft mode is not only explained by temperature. I think this statement is not understandable at this point either. It may be best to either leave out this statement or say that a finite temperature effects are observed."

We have indeed meant that the frequency of the mode has a powerlaw scaling near the critical point and have clarified this now.

We have modified the last sentences of the introduction (lines 76 to 80) to explain clearer the achieved results of the paper. In particular, we highlighted, which results were experimental and which numerical. The modified text now reads:

"We show using numerical calculations that the soft mode frequency exhibits a powerlaw scaling behavior in the vicinity of the critical point, where the system becomes superlubric. The experimental spectroscopic measurements show a small reduction in the frequency of the soft mode. The non-vanishing frequency of the sliding mode is due to the finite temperature exciting nonlinear dynamics. Additionally, the experimentally observed order parameter agrees with numerical results, which also indicate a power law scaling in the vicinity of the critical point"

The numerical simulations together with Fourier analysis of the ion's motion, indicated that the experimentally observed frequencies of the soft mode were determined by the finite temperature in our system. This is discussed later in the manuscript, line 276, where we now tried to write this more clearly. "...we conducted molecular dynamics simulations of the unperturbed crystal, i.e. without laser excitation, at finite temperatures (see Methods). From the ion trajectories the vibrational spectrum of the crystal was extracted using a Fourier transformation.... We attribute the deviation of the observed frequencies near the transition point to the increasing contribution of the nonlinearities of the PN potential, which is shown in Fig. 3b."

"line 85 and following: it is not clear to me how relevant the inhomogeneity is in all these experiments. I think this should be addressed in detail."

The effect of inhomogeneity was investigated, but perhaps we did not stress this enough in the manuscript. We have evaluated the effect of inhomogeneity by performing calculations of the dispersion relation for different system sizes with 30, 60 and 90 ions since the larger the crystal the smaller the inhomogeneity. The scaling is essentially unaffected by the variation of the number of ions (Fig. 4). We have also investigated numerically crystals with periodic boundary conditions (PBC) in the axial direction. This effectively removes the inhomogeneity. Comparing the depinning force (force needed to translate a kink by a lattice site) in a harmonically trapped crystal and the crystal with PBC, revealed that the effect of inhomogeneity is only to shift the critical force by some small offset without modifying the scaling behaviour (Fig. 2e). The reason why our results are not influenced by the level of inhomogeneity is that we focus on the phenomena of the onset of sliding, which is dictated by the localized defect. Certainly, the inhomogeneity will become relevant if the rows are forced to move with respect to one another by many lattice sites.

In the revised version of the manuscript, we have stated this more clearly and in particular added the following sentence to the discussion section:

"Typically, trapped ions are confined in harmonic potentials which lead to an inhomogeneous ion spacing. Our results show that the inhomogeneity due to the boundary conditions and the system size influence our model system only slightly. This is due to the fact, that we focus on the phenomena of the onset of sliding, which is dictated by the localized lattice defect."

"line 95: "Its frequency depends on .." The previous sentence just stated that its frequency is zero. This bears

some explanation for instance that it is not 0 in a finite size system, etc."

We removed the sentence around line 95, because we realised that it would be confusing at this point. It is explained clearer in the section "Soft mode and critical scaling".

"Line 173: I do not understand the sense in the procedure "Then we apply both forces in the same direction ..." This makes no sense to me as the relative positions of the two layers will not change and hence this force should not do anything other than to accelerate both crystals together."

This is correct. Our intention was to achieve a situation, where the influence of the corrugation potential of the FK-model can be "switched off" in our self-assembled system, and where the unperturbed coordinates of the ion chain can be extracted. We have clarified how we calculate the hull function by improving the sentence around line 170. It reads now:

"In the classical FK model, the hull function $z_j(z_{j,0})$ is defined as the coordinate of a particle j under the influence of a static corrugation potential in relation to its unperturbed coordinate $z_{j,0}$ without the underlying lattice. For the self-assembled system of two interacting atomic chains, the corrugation potential is given by the Coulomb potential of the 2nd row of ions and thus cannot be switched off. In order to calculate the hull function in such a system, we first apply opposite forces $\pm F/2$ to each row of the crystal to obtain z_j . Then we apply the force F to both rows in the same direction, moving the lattice along with the ion to obtain $z_{j,0}$, i.e. the equilibrium position of the ions in the harmonic trapping potential without any influence of the underlying lattice. We implement this principle in our numerical simulations and show the result in Fig. 2a."

"line 183:

fig 2a: what do the uncertainties mean? What is a ? The lattice spacing? Should it have units?"

We realize that the character " a " and α look similar in the given fonts. We have clarified this now in the caption of Fig. 2, stating that α is the aspect ratio of the radial to axial trapping frequency. Its uncertainty is determined by the uncertainties of our secular frequency measurements in our ion trap, which are typically around 100 Hz, see Methods.

"fig 2c: why is the hull function only displayed from -0.04 till 0.04, should it not go from -0.5 to 0.5? I guess there is nothing interesting happening there. But if nothing interesting is happening there, then I start to wonder whether the present system is not in some weird limit of the Frenkel-Kontorova model. Finally, is this a measurement?"

The data in Fig. 2c (now Fig. 2a) are from simulations. We stated that clearly now in the caption. The motion of ions in this system with locally broken commensurability is quite interesting. While the ions move very little, the motion of the local defect (i.e. the higher charge density) is much larger. The finite size of our system does not allow to apply larger forces, otherwise the defect is lost into the open end of the crystal. One possibility would be to go to larger crystals or, in an ideal case, to a ring trap.

"fig 2e: I am guessing that those come from simulations. This should be made clear right from the beginning. It left me pondering how the hell this was measured for quite some time. The implicit statement the following sentence is not sufficient.

fig 4: indicate that those are numerical results."

The referee is correct in that the results in Fig. 2e come from numerical simulations. We have now explicitly stated in the captions of all figures the nature of the results.

We thank the referee for the careful and helpful review of our paper.

Reply to reviewer #2:

We thank the referee for critical reading of our manuscript and his helpful comments. In the following paragraphs we answer his questions in detail.

(Line numbers refer to the originally submitted manuscript)

Note: we have reordered Fig. 2a,b,c to be more coherent with the main text.

“This combined experimental and numerical study reports on the frictional properties of two interacting and incommensurate layers of an ion Coulomb crystal. In the case considered here, the incommensurability is introduced by a single topological defect, i.e. a kink. When the electrostatic coupling between the layers is gradually decreased, a soft vibrational mode is observed which corresponds the disappearance of static friction in agreement with an Aubry transition. Both, the soft mode and the structural order parameter display a power-law close to the critical point. These findings are corroborated by numerical simulations.

The Aubry transition is an intriguing phenomenon which has been rigorously, i.e. mathematically proven for infinite, one-dimensional systems. When reading the paper, I first thought, that the authors have performed measurements on a two-dimensional system (because they repeatedly refer to layers), however, then I realized, that they only considered the case of two chains of interacting particles. I understand, that their work adds additional information beyond a recent similar study of Bylinskii et al., but I do not consider the results of this manuscript not as a substantial improvement compared to the other one. Therefore, I do not recommend it for Nat. Comm. but suggest to submit this to another, more specialized journal.”

The referee is right in that the system at first sight looks like a 2D system and that it has the interesting situation that the ions in both chains do have two degrees of freedom to move in this self-assembled system. One of the objectives of the paper is to show that the system nevertheless exhibits a one dimensional finite Aubry phase transition, which is a second-order phase transition. This is what gives additional value to our work and separates it from previously published work.

We carry out all of the calculations on the full three dimensional Coulomb crystal system without mapping to one dimensional models (e.g. FK model). The results strongly support our original intuition, that the system exhibits a one dimensional Aubry transition.

In addition, substantially different compared to Bylinskii et. al. is that we have a system featuring back action.

Some of the questions the referee put out to be answered are highly interesting and cannot be answered easily. We hope that our work will raise interest at theorists to analyse such a system.

“Below, please find my questions and remarks:

1. As the authors certainly know, the nature of the Aubry transition changes from continuous two first-order when going from 1D to 2D. When looking at their experimental situation (Fig.1a), my feeling is, that they are somewhere in between. What other evidence the authors have, that a 1D description is valid?”

All of the calculations were performed on a system with full dimensionality. The results reveal signatures of the finite 1D Aubry phase transition. These are – sliding mode becoming soft at a critical point, symmetry breaking with continuous change of the order parameter, scaling of the restoring force and the frequency of the soft mode. These results are presented in figures 2, 3 and 4. This evidence is essentially empirical rather than deductive. In the future, it would be interesting to develop a more formal treatment deriving the mathematical model for this transition, but this is beyond the scope of the present work.

We believe that a two dimensional Aubry transition can be investigated in a similar way as presented in our work. This would require 1) creating large planar crystal, 2) diabatically inducing a transition to two planes thereby creating lattice with defects 3) varying the distances between these planes and 4) observing the ensuing effect on the dynamics. This would be a very interesting extension of the present work and we feel that it would be of benefit to mention this possibility in the final section of the main text of the manuscript. We have added the following sentence with two new references around line 343

“Moreover, the ideas presented in this work may be used to investigate the currently poorly understood Aubry transition in two dimensional systems [Phys Rev B, 92, 134306 (2015)], since ion traps can trap large three dimensional Coulomb crystal composed of regular two dimensional layers [Science 282, 1290 (1998)]”

2. Compared to previous work where the Aubry transition was demonstrated in a system of 5 ions only, here the number of ions is somewhat larger but it is still far away from the infinite case considered by Aubry. How does the finiteness of the system affect the way how the soft mode develops? How Fig.3a would vary if one, lets say doubles the chain length?

The behavior of the soft mode over a larger span of α was calculated but not shown in the original manuscript. We added this now in *Supplementary Figure 3* showing 30 and 60 ions in the Supplementary Part.

We have also investigated the effect of the number of ions on the frequency of the soft mode numerically at zero temperature close to the critical point. The results were presented for a 30, 60 and 90 ion chain in the loglog plot in Fig. 4. The scaling is not affected by the number of ions as the defect remains a localized structure.

3. The authors say, that the axial harmonic confinement of the ion trap leads to an inhomogeneous particle spacing but that the central part of the crystals exhibits a (quasi)-periodic lattice. What do the authors mean by the word “quasi”? I assume, that they do not refer to quasiperiodic order, but that the particle distance continuously changes. Then, however, I do not really understand the meaning of incommensurability.

We agree that the term (quasi)-periodic lattice was not a conventional term in the present context. We have changed the text to “slowly varying lattice spacing” (line 91).

In Bylinskii et al [23], the chain is also inhomogeneous but it slides on a homogeneous (perfectly periodic) lattice. There, the ion periodicity is not matching perfectly but only locally, which is justifiable if the inhomogeneity is varying slowly.

In our case, both upper and lower row of the crystal have the same level of inhomogeneity since they are in the same harmonic confinement. The incommensurability is then determined by the relative distances of the upper chain with respect to the lower chain. We change the incommensurability by introducing an extra ion in one of the rows i.e. by creating a structural defect. Without the presence of the local defect, the relative ion spacing between upper to lower rows would be commensurate even under the presence of the inhomogeneity.

4. The presence of a soft mode is indeed a signature of the Aubry transition. In contrast to Aubry, who considered an elastically coupled chain of particles in a strictly periodic substrate potential, here the authors consider the situation of two interacting chains of particles. Similar to my question 1, how such soft modes are influenced by considering also further vertical couplings between the particles?

In the work presented in this manuscript, we have evaluated the frequency dependence of the soft mode by diagonalizing the full Hessian matrix thereby including the effect of the vertical motion of the ions and long range couplings between the ions. We have not investigated systematically how imposing various restrictions into the system e.g. allowing only horizontal motion, switching off long range interactions etc...

would affect our system. This could be an interesting and significant undertaking that we leave for future explorations.

To our knowledge there is little previous theoretical investigation on the effect of the coupling of the horizontal and vertical degrees of freedom on the Aubry type transition. One such work is by Einax and Schultz [Phys Rev E, 70, 046113 (2004)], which considers two long chains interacting via a Lennard-Jones-like potential. In that paper, they observe an Aubry type transition, but focus primarily on the critical behaviour of the hull functions.

We thank the referee for their careful reading and helpful comments on the manuscript.

Reply to reviewer #3:

“The manuscript reports a study of stick-slip friction using a two-layered ion crystal in a zig-zag arrangement. The idea to translate two ion chains against each other to implement the Frenkel-Kontorova model is ingenious. The paper reports interesting physics in a novel system, and sheds new light on microscopic friction models in finite systems. As such, the paper is clearly suited for Nature Communications.”

The referee correctly states that the aim of the paper is to study stick-slip motion using a two-layered ion crystal without resorting to externally applied external periodic field. We are pleased that the referee finds the manuscript interesting and novel, and recommends the suitability for Nature Communications. In the following we answer his or her questions and comments.

(Line numbers refer to the originally submitted manuscript)

Note: we have reordered Fig. 2a,b,c to be more coherent with the main text.

“However, I think that the presentation could be substantially improved. The current version of the manuscript is difficult to follow for even for the specialist, let alone for the general reader. The manuscript jumps back and forth between theoretical expectations, experimental results, and numerical modeling, and does not always make clear which is which.”

We agree with the referee that the original version of the manuscript was confusing in this matter. It was certainly not our intention and this is an oversight on our part. We improved it now and clearly state throughout the whole manuscript where experimental and numerical results are used, e.g. line 176 reads now:

“This is visible in Fig. 2b, which shows photos of experimental realizations of 30 ion crystals at different α . They are compared to images obtained from numerical simulations at $T = 1$ mK, shown in Fig. 2c.”

We have modified the last section of the introduction (lines 68 to 81), where we now state more precisely what where the findings of our study, clearly separating which results were numerical and which experimental:

“We show using numerical calculations that the soft mode frequency exhibits a power law scaling behaviour in the vicinity of the critical point, where the system becomes superlubric. The experimental spectroscopic measurements show a small reduction in the frequency of the soft mode. The non-vanishing frequency of the sliding mode is due to the finite temperature exciting nonlinear dynamics. Additionally, the experimentally measured order parameter agrees with numerical results, which also exhibit a power law scaling in the vicinity of the critical point.”

Also in the captions of all figures the nature of the results is explicitly stated now.

“Furthermore, and perhaps relatedly, some of the main claims of the abstract and manuscript are misleading, and appear not to be substantiated by experimental results. For instance, the abstract states: “We show that local defects in the periodicity of the atomic layers cause a transition from sticking to sliding with Aubry type signatures.” To the best of my understanding of the manuscript, the authors do not show a comparison of friction with and without local defects, as this sentence seems to imply, but only that there is a transition in the distribution of ion positions in the presence of a defect when the ratio of trap frequencies is varied. Perhaps the authors meant something along the lines: “We show that in a system with local defects resulting in incommensurate layers, there is a transition from sticking to sliding with Aubry type signatures.”

We agree that the referee’s suggested sentence “We show that in a system with local defects resulting in incommensurate layers, there is a transition from sticking to sliding with Aubry type signatures.” reflects better the aim and results of our work. We have modified this sentence (line 17-18) according to the referee’s suggestion.

In fact, we did consider friction in the same system but without defects i.e. commensurate lattice, though this was only a minor part of the paper. We have found that the force needed to shift the two lattices with respect to one another is an order of magnitude greater, if the two lattices are commensurate. For commensurate lattices we performed the calculation using the Prandtl-Tomlinson model. The results are discussed in the lines 251 to 258 and the details of the calculation are in the Supplementary Part.

We have now added a third graph to Fig. 2e showing the friction force of an ideal crystal, which is roughly an order of magnitude higher near the transition. This was previously only mentioned in the caption of this figure.

Furthermore, a clear distinction is that ideal crystals without defect exhibit no soft mode. Here, the frequency of the shear mode is always finite and much larger than e.g. the breathing mode of the system. Only when a local defect is present, the soft mode and therefore the transition exists. We have added now a sentence in the first paragraph of the section “Soft mode and critical scaling” that stresses this point.

“More importantly, the abstract claims: “We demonstrate spectroscopic frequency measurements of the vibrational soft mode driving this transition and ... Both [one of them being the vibrational mode] exhibit critical scaling near the transition point.” Now the experimental data in Fig. 3a only show a slow weak and linear reduction of the “soft-mode” frequency up to the critical point, with nothing particular happening at the presumed critical point. In numerical simulations also shown in this figure, this absence of an observation is explained by the finite, and relatively large, temperature of the system. Nonetheless, the claim of “critical scaling near the transition point” is clearly not borne out at all by the experimental data. (There is even no convincing mode softening at the critical point.) All claims of critical scaling derive purely from numerical simulations in Figures 3 and 4. So clearly the abstract and main text need to be reformulated to make clear what

was, and what was not, observed. Critical scaling was not observed in the experiment, only in numerical simulations. The caption of Figure 4 should also state that this is a simulation.

Throughout the rest of the manuscript there are similar confusing jumps back and forth between experiment, numerical simulations, and theoretical expectations from the model. Many of the further reaching claims, such as the ones discussed above or the friction force in Fig. 2e, are only supported by

numerical simulations, not by the experimental data.”

We have modified the abstract to make it clear, which conclusions come from experiments and which from the numerical simulation. The sentences in the lines 18 to 20 in the original manuscript now read: “We demonstrate spectroscopic frequency measurements of the vibrational soft mode driving this transition and a measurement of the order parameter, which characterizes the symmetry breaking. We show numerically that both exhibit critical scaling near the transition point.” Similarly in the main text, in the introduction (lines 68-82), we summarize the results clearly differentiating experimental results and numerical results. In all figure captions, we now emphasized whether the results come from simulations or experiments.

“The manuscript also does not explain how the experiment was actually performed. How and with what time dependence was the displacement force applied, or are the actual data averages over many different fixed forces? This should be stated in a main text. (p. 5, the caption to Figure 1 states “The chains can move against each other, e.g. using laser light forces.” Why “e.g.” here, what was actually done?) Was the displacement force for Figure 2 applied with a sinusoidal time dependence? If so, would the ions not spend more time at the turning points, and therefore a gap in the position distribution might appear that mimics the gap of the Aubry transition?”

Fig. 2a and b (now 2b and c) display pictures of the equilibrium positions of ion Coulomb crystals at different aspect ratios of the ion trap α averaged over 700 ms exposure time. No force is applied the motion here is purely temperature driven. At finite temperatures the ion positions are smeared out depending on how close the system is to the critical point. The gapped ion positions are explained by appearance of multiple stable crystal configurations after the crossing of the critical point. The thermal energy of the ions allows to overcome the PN barriers and thus makes the various configurations visible.

However, the laser force used to excite the sliding mode to measure the dispersion of this mode, as given in Fig. 3, had indeed the sinusoidal time dependence.

Most of the experimental details are given in the Methods section of the manuscript and in the captions of the figures. However, we agree with the referee that the manuscript would be easier to follow, if we give a more complete description of the experimental setup and protocol in the main text. We have made it clearer now in the main text, from where and how the data were obtained and added sentences to explain e.g.: “Experimentally we use differential laser light forces and resonantly excite the vibrational modes of the ion Coulomb crystal by sinusoidal intensity modulation”.

“Why do the ion positions not appear to be smeared out in Fig. 2a (I) if this represents the sliding phase? Is the amplitude of the sliding motion too small to see in the picture 2a(I), or was the amplitude changed between (I) and (VI)? Does Fig. 2C represent experimental data or merely numerical simulations? If the latter is the case, this should be pointed out in the figure caption.”

The previous works on the finite Aubry transition [Braiman et al, 1990; Benassi et al 2011, Pruttivarasin 2011; Bylinskii 2016] have established that in finite system the nature of the Aubry transition changes. Most notably, a truly sliding Aubry phase exists only in infinite system. The finite Aubry transition is a symmetry breaking phase transition and these previous works identified the symmetric state as the sliding regime and the symmetry broken state as the pinned regime. However, strictly speaking both states in finite systems are pinned – the sliding mode frequency is only zero at a critical point, see also Fig. 3a. For this reason, we see that the blurring of the ion position is only large in the vicinity of the transition i.e. Fig. 2a (III) (now Fig. 2b (III)).

Fig. 2c (now Fig. 2a) represents data obtained using numerical simulations. We have now clearly stated that in the captions.

“Overall, the manuscript should be improved to make the experimental procedure and achievements clearer. My suggestion would be to first discuss the experimental results, and then separately their theoretical interpretation, so that it is clearer what has been experimentally achieved, and how. As far as I can tell, the main experimental achievements are 1) the opening of the gap in position or order parameter as presented in Figure 2d, and the observation of a modest softening of the vibrational mode as presented by the red circles in Figure 3a. Since according to calculations one should observe a real dip in the transition frequency at the critical point, it should also be discussed what limits the temperature, and if there are prospects to repeat the measurement at lower temperature. The information presented in all other figures appears to be coming exclusively from numerical calculation, but it is not always clear, as in Figure 2c.”

We agree with the referee that the manuscript would benefit from stating more clearly, which results were obtained using experiments and which using numerical calculations. We have improved this now in the main text as well in all the figure captions.

Regarding the structure, the manuscript focuses sequentially on the different aspects of the critical phenomena of the finite size Aubry transition, that is we 1) introduce the system 2) investigate the order parameter and the depinning force 3) investigate the mode frequency and 4) draw conclusions. The numerical simulations and experiments are complementary and the concurrent application of these various methodologies was essential in the development of the present work.

For this reason, we would prefer to maintain the original order of presentation but strongly and clearly emphasize the distinction and interrelation between the experimental and numerical results.

E.g. we have added the following sentences to the end of the introduction, which lay out the structure of the manuscript: “The manuscript is structured as follows: in the first section we introduce our experiment and model system. In the next section, we investigate the structural features of an Aubry-type transition - the symmetry breaking, the order parameter and the hull function. We then introduce the spectroscopic findings on the soft mode. Finally, we discuss our results and prospects of our model system.”

We have reordered the graphs in Fig. 2 (a,b,c) to be more coherent with the main text.

In order to clarify the interrelation between the shown data, we e.g. explain now when discussing the order parameter of Fig. 2d: “The experimental data in Fig. 2d (red circles) are extracted from CCD images, some of which are shown in Fig. 2b.”

Regarding the temperature in our system, we are currently limited at the Doppler temperature of the $S_{1/2} \rightarrow P_{1/2}$ transition needed to trap and cool the ion Coulomb crystals. We have precised the details of this in the Methods section now.

One possibility to further reduce this temperature is to implement a second-stage cooling.

We addressed this point in the “Discussion” at the very end. “In the future, the experiments can be improved by further cooling to the μK regime using narrow transitions [32] or dark resonances [43,44], ...”

In our case, we could implement a second-stage cooling on the intercombination line of In^+ ions to lower the temperature to some tens of μK .

“Overall, I quite like this work and the idea behind it, and I recommend publication in Nature Communications. However, I think that the authors could significantly strengthen the impact of their work by restructuring their manuscript to improve its readability, and by clearly stating claims that are substantiated by experimental data.”

We hope that in our revised version the aims and the achievements are stated much more clearly. In particular, we believe that it is now clear from the abstract, introduction as well in all figures, which results were obtained experimentally and which using simulation and numerical calculations.

We thank the referee for their careful reading and helpful suggestions for the manuscript.

Reply to reviewer #3:

“The authors have responded adequately to the referees, and I find the manuscript substantially improved, and much clearer. There are only a few remaining small issues:”

We are happy to hear that our changes helped clarify our manuscript. We will now address the remaining issues.

“i) Upon re-reading the title, “self-assembled system” seems a little strange, after all, the ions are trapped in an external trap, and the crystal does not “self-assemble”, it at most “self-organizes” under the influence of laser cooling. However, I don’t feel strongly about this issue.”

In the manuscript, we have used the words "self-assembled" and "self-organized" interchangeably. However, we agree with the referee that "self-organized" is a more appropriate term in the context of this work, and therefore we have changed "self-assembled" to "self-organized" throughout the article.

“ii) In their response to me regarding what is shown in Figure 2, the authors reply:” Fig. 2a and b (now 2b and c) display pictures of the equilibrium positions of ion Coulomb crystals at different aspect ratios of the ion trap α averaged over 700 ms exposure time. No force is applied the motion here is purely temperature driven. At finite temperatures the ion positions are smeared out depending on how close the system is to the critical point. The gapped ion positions are explained by appearance of multiple stable crystal configurations after the crossing of the critical point. The thermal energy of the ions allows to overcome the PN barriers and thus makes the various configurations visible.” This response is very clear, but the Figure caption of Figure 2 is not, i.e. it is not clear that it is the thermal energy rather than an applied force that makes the various configurations observable. I recommend that the authors modify the caption along the lines of the above response.”

To clearly state in the caption, that the thermal motion is responsible for the observed effects, we have rewritten the caption of Fig. 2b. It now reads:

“Photos of experimentally observed 30 ion crystal configurations at different aspect ratios $\alpha = \frac{\omega_{\text{rad}}}{\omega_{\text{ax}}}$ from 5.29 to 7.21, as indicated in the grey bar. Relative errors are less than 1%. The exposure time is 700 ms. Laser 1 continuously cools the ions. No force is applied. The blurring of the ion positions near the sliding transition and the appearance of multiple configurations above it are due to thermal excitations. The scale bar is 16.5 μm .”

”

“iii) Line 194: “The experimentally and numerically observed critical corrugation parameter ...is smaller than 1, because for simplicity the interaction energies were calculated only for zigzag.” I understand how this approximation can affect the calculation, but how can it affect the experimentally observed critical corrugation parameter?”

From a practical perspective, the control parameter for both numerical and physical experiments is the trap aspect ratio α . The critical point obtained from both the numerical and experimental data is $\alpha_c \approx 6.4$.

As η is neither a direct parameter of the simulation, nor measurable in the experiment, we have mapped α to η in the same way for both experimental and numerical sets of results. This explains why this approximation affects both experimental and numerical results identically. The details of how we related the aspect ratio α to the ratio of the inter-row and intra-row interaction energies are given in the last paragraph of the section "Experimental system".

"This is an interesting and novel microscopic study of friction that is of interest to the general audience. In summary, I recommend publication of this manuscript in Nature Communications with minor revisions."

We thank the referee for their review our paper and recommendation for publication.